# Simultaneous Analysis and Dietary Exposure Risk Assessment of Fomesafen, Clomazone, Clethodim and Its Two Metabolites in Soybean Ecosystem

**DOI:** 10.3390/ijerph17061951

**Published:** 2020-03-17

**Authors:** Kyongjin Pang, Jiye Hu

**Affiliations:** School of Chemistry and Biological Engineering, University of Science Technology Beijing, Beijing 100083, China; banggyongjin@163.com

**Keywords:** fomesafen, clomazone, clethodim and its metabolites, dietary exposure risk assessment, LC-MS/MS

## Abstract

A commercial formulation, 37% dispersible oil suspension (DOS) (fomesafen, clomazone, and clethodim), is being registered in China to control annual or perennial weeds in soybean fields. In this paper, a liquid chromatography tandem mass spectrometry method with QuEChERS (quick, easy, cheap, effective, rugged, and safe) sample preparation was developed for the simultaneous determination of fomesafen, clomazone, clethodim, and its two metabolites (CSO and CSO_2_) in soybean, green soybean, and soybean straw samples. The mean recoveries of our developed method for the five analytes in three matrices were ranged from 71% to 116% with relative standard deviations (RSDs) less than 12.6%. The limits of quantification (LOQs) were 0.01 mg/kg in soybean, 0.01 mg/kg in green soybean, and 0.02 mg/kg in soybean straw while the limits of detection (LODs) ranged from 0.018 to 0.125 μg/kg for these five analytes. The highest final residual amount of CSO_2_ in green soybean samples (0.015 mg/kg) appeared in Anhui, and the highest in soybean straw samples was 0.029 mg/kg in Guangxi, whilst the terminal residues of fomesafen, clomazone, clethodim and CSO were lower than LOQs (0.01 mg/kg) in all samples. Furthermore, these terminal residues were all lower than the maximum residue limits (MRLs) set by China (0.1 mg/kg for fomesafen and clethodim, 0.05 mg/kg for clomazone) at harvest. Additional chronic dietary risk was evaluated using a risk quotients (RQs) method based on Chinese dietary habits. The chronic dietary exposure risk quotients were 4.3 for fomesafen, 0.12 for clomazone, and 19.3 for clethodim, respectively, which were significantly lower than 100. These results demonstrated that the dietary exposure risk of fomesafen, clomazone, and clethodim used in soybean according to good agricultural practices (GAP) was acceptable and would not pose an unacceptable health risk to Chinese consumers. These results not only offer insight with respect to the analytes, but also contribute to environmental protection and food safety.

## 1. Introduction

Soybean is one of the most important and widely grown crops in the world. For example, the annual productions of soybean around the world were as high as 13.15 million tons in China, 7.7 million tons in Canada, 114.59 million tons in Brazil and 119.5 million tons in USA [1], respectively. Not only the huge production, but the corresponding consumption were closely related to the use of pesticides. These pesticides are intentionally toxic towards target and non-target organisms, wherein some pesticides have been suspected to have negative effects in human health risk [2,3]. As a commercial formulation, 37% dispersible oil suspension (DOS) (fomesafen, clomazone, and clethodim) is being registered in China, and it is expected to be widely used in soybean fields to control annual or perennial weeds in the near future. The chemical structures of fomesafen, clomazone, clethodim, and its two metabolites are shown in Figure 1.

Fomesafen has been used for post-emergence weed control in soybean [4], tomato [5], wheat [6], strawberry [7], and corn fields [8]. However, fomesafen has demonstrates persistence in various soils [4,7] due to its anionic character and water-solubility (120 mg/L) [9]. In addition, fomesafen and its residue exhibit phytotoxicity during crop rotation and accumulation in soil [4]. Meanwhile, the reproductive toxicity of fomesafen in freshwater snails and hepatic uroporphyrin in mice are also reported [10].

Clomazone which belongs to the isoxazolidinone family [11] has been used for the control of annual grasses and broadleaf weeds in cotton [12], teas, potatoes, rice [13], squash, cassava, soybeans [14,15], sweet tobacco [16], wheat [17] and a variety of other vegetable crops [18]. Clomazone which belongs to toxicological class II inhibits the chloroplastic isoprenoid pathway in higher plants [19]. Clomazone is highly water-soluble (1100 mg/L), resistant to hydrolysis under a wide range of pH values and weakly absorbed in soil. Thus, it may cause groundwater contamination [20].

Clethodim belongs to the family of cyclohexanedione oxime herbicides, making it a selective post-emergence herbicide [21]. It has been widely used for the cultivation of soybean [22], peanut, maize, and rape [23] in European countries and China [24,25]. Similar to other cyclohexanedione oxime herbicides, clethodim is highly water-soluble (5450 mg/L) and poorly adsorbed in soil (Log P = 2.5). Hence, it may move into aqueous system and become a potential contaminant [21,24,25]. Moreover, clethodim is rapidly photolyzed to some photoproducts, such as sulfoxides and dechloroallyloxy derivatives which can remain in the aqueous media for a longer time compared with the parent clethodim [25,26,27]. Thus, the Joint Meeting on Pesticide Residues (JMPR) recommended that the residue definition of clethodim for compliance with maximum residue levels (MRLs) and for the estimation of dietary intake should be the sum of clethodim, CSO (clethodim sulfoxide) and CSO_2_ (clethodim sulfone), expressed as clethodim [25].

Several analytical methods were conducted to measure the fomesafen, clomazon, or clethodim residues in some different matrices. The residues of fomesafen were detected in tomato [5] and bovine milk [28] by liquid-liquid extraction or QuEChERS (quick, easy, cheap, effective, rugged, and safe) sample preparation and LC-MS/MS (liquid chromatography tandem mass spectrometry). Moreover, the residues of fomesafen in corn and soil were also detected by methanol/hydrochloric acid of liquid-liquid extraction and high-performance liquid chromatography with diode-array detector (HPLC-DAD) [8]. The cyclic voltammetry and square wave voltammetry method were also employed to determine the residues of fomesafen in juice [29]. The literature also reported that the residues of clomazone were detected in soybean and soil using the HPLC-DAD method [14,15] and in bovine milk by LC-MS/MS [28]. As for clethodim and its metabolites, a method was developed for the determination in agricultural products such as radish, tomato, onion, sweet potato, carrot, cabbage, and lettuce by liquid-liquid extraction [30] and tobacco by QuEChERS sample preparation [31].

Up to now, no method has been found for the simultaneous analysis of these five analytes in soybean ecosystem. Therefore, the development of an effective method for the simultaneous analysis of fomesafen, clomazone, clethodim, and two metabolites in the soybean ecosystem is one of the major important issues in food analysis. In addition, the chronic dietary risk evaluation of fomesafen, clomazone and clethodim for Chinese consumers has not been reported in the public literature.

One aim of the present work is to develop an accurate, reproducible, and sensitive method for the simultaneous determination of fomesafen, clomazone, clethodim, and its two metabolites CSO and CSO_2_ in soybean, green soybean and soybean straw samples based on QuEChERS sample preparation [32] and UPLC-ESI-MS/MS (Agilent Technologies, Santa clara, CA, USA). The other aim of the present work is to investigate the residue distributions of the three herbicides and two metabolites under the open field conditions using the established method and is to assess dietary risk probability of three pesticides in soybean and green soybean based on field trial data and relevant toxicological parameters.

## 2. Materials and Methods

### 2.1. Reagents and Chemicals

Five standards of fomesafen (97.0%), clomazone (98.2%), clethodim (99.0%), CSO (94.1%), and CSO_2_ (94.1%) were purchased from National Center for Quality Supervision and Testing of Pesticides (Shenyang, China). Further, the 37% fomesafen-clomazone-clethodim dispersible oil suspension (37% DOS) containing 11% fomesafen, 21% clomazone and 5% clethodim was provided by Shandong Ludedi biotechnology Co., Ltd. (Shandong, China). Acetonitrile and formic acid were purchased from Fisher Scientific (Far Lawn, NJ, USA). Sodium chloride (NaCl), anhydrous magnesium sulfate (MgSO_4_), and acetic acid of analytical grade were purchased from Beijing Chemical Reagents Company (Beijing, China). Primary and Secondary Amine (PSA, 40–60 μm), Florisil, graphitized carbon black (GCB, 40–60 μm), multi-walled carbon nanotubes (MWCNT, length of 10–30 µm, diameter of 10–20 nm) and syringe filter (polytetrafluoroethylene, PTEF, 0.22 μm) were obtained from Tianjin Bonna-Agela Technologies (Tianjin, China).

### 2.2. Solution 0reparation

The standard stock solutions (1000 mg/L approximately) were prepared by dissolving 25.8 mg fomesafen, 25.5 mg clomazone, 25.3 mg clethodim, and 26.6 mg CSO and CSO_2_ into a 25 mL brown bottle with HPLC-grade acetonitrile, separately. The mixed standard working solution (50, 5, 0.5 mg/L) of fomesafen, clomazone, clethodim, CSO and CSO_2_ was prepared by diluting each standard stock solution (1000 mg/L) with HPLC-grade acetonitrile. The matrix-matched standard calibrations of the five analytes were sequentially 1, 0.5, 0.1, 0.05, 0.001, and 0.0005 mg/L in soybean, green soybean and soybean straw by the sequential diluted with blank matrices, respectively. All standard solutions were freshly prepared, filtered through 0.22 µm nylon membrane filters, and kept in the dark at 4 °C.

### 2.3. Sample Preparation by QuEChERS

The soybean (5 g) samples were weighed into a 50 mL PTFE centrifuge tube, separately. 10 mL acetonitrile with 1% (*v:v*) formic acid was added. After shaking vigorously for 1 min, it was further shaken in an air bath at 25 °C for 20 min. Then, anhydrous NaCl (1 g) and anhydrous MgSO_4_ (3 g) were added and vortexed vigorously for 1 min. After centrifugation at 4000 rpm (revolution per minute) for 3 min, 1 mL of the supernatant was transferred into 5 mL centrifuge tube which contained the dispersive solid-phase extraction sorbents of 150 mg MgSO_4_ and 50 mg C18. After shaking vigorously for 1 min, it was centrifuged for 3 min at 10,000 rpm. Then, the supernatant was filtered with a 0.22-µm nylon membrane filters for LC–ESI-MS/MS analysis.

The green soybean and soybean straw samples were prepared with similar procedures compared with soybean samples. The green soybean (5 g) and soybean straw (2.5 g) samples were weighed into a 50 mL PTFE centrifuge tube, separately. Then, the 10 mL acetonitrile with 1% (*v:v*) formic acid was added. After shaking vigorously for 1 min, it was further shaken in an air bath at 25 °C for 20 min. Anhydrous NaCl (1 g) and anhydrous MgSO_4_ (3 g) were added and vortexed vigorously for 1 min. After centrifugation at 4000 rpm for 3 min, 1 mL of the supernatant was transferred into a 5.0 mL centrifuge tube, which contained the dispersive solid-phase extraction sorbents 150 mg MgSO_4_, 50 mg C18, and 5 mg MWCNT, and was shaken vigorously for 1 min. Then, it was centrifuged for 3 min at 10,000 rpm. The supernatant was filtered with 0.22 µm nylon membrane filters for UPLC–ESI-MS/MS analysis.

### 2.4. UPLC-ESI-MS/MS Condition

The residue determination of fomesafen, clomazone, clethodim, CSO and CSO_2_ in soybean, green soybean and soybean straw were performed on Agilent 1260 infinity LC system (Agilent Technologies, CA, USA) with an Agilent Poroshell 120 EC-C18 (50 × 3.0 mm i.d., 2.7 μm). The LC system was coupled on-line to Agilent 6460 triple quadrupole mass spectrometer (Agilent Technologies, Santa clara, CA, USA) equipped with an electro-spray ionization source (ESI). Five compounds were separated with an isocratic mobile phase consisting of 75% acetonitrile and 25% water containing of 10 mM ammonium acetate. The flow rate was 0.35 mL/min. The injection volume for all samples was 5 µL and the column temperature was 25 °C. Other parameters including the drying gas (N_2_) temperature of 300 °C and flow rate of 11 L/min, the capillary voltage of 4 kV in positive mode and 3.5 kV in negative mode, the nebulizing gas (N_2_) pressure of 35 psi were performed to analyze these five substances.

### 2.5. Method Validation

To assess the validation of our method, recovery experiments were carried out in the soybean, green soybean and soybean straw samples. The mixed standard working solutions of fomesafen, clomazone, clethodim, CSO and CSO_2_ were fortified to the three control matrices. The spiked levels were chosen as 0.01, 0.1, 1 mg/kg for both soybean and green soybean, and 0.02, 0.1, 1 mg/kg for soybean straw, respectively. Each fortification was carried out with five replications (*n* = 5). These samples were processed and analyzed according to the above procedure. Meanwhile, non-spiked samples were also analyzed to check interference of the matrix.

To assess the influence of co-extracted impurity substances on analytical signal intensity during the UPLC-MS/MS detection, matrix effects (ME) was evaluated by comparing the slope of the matrix-matched calibration curves to the slope of the calibration curves in acetonitrile. ME (%) = (slopes of the matrix-matched calibration curve-slopes of the calibration curves in acetonitrile)/slopes of the calibration curves in acetonitrile) × 100.

### 2.6. Field Trials

The field trials of the residue experiments were carried out at six sites in China in the year of 2018 according to the Standard Operating Procedures on Pesticide Registration Residue Field Trials compiled by the Institute for the Control of Agrochemicals, Ministry of Agriculture and Guideline on Pesticide Residue Trials (The Ministry of Agriculture of China, 2018). Figure 2 illustrated the six sites of field trial in China in 2018. The six sites in open field were as follows: Harbin (128°45′ E, 45°05′ N, Heilongjiang province, Northeast of China), Ulanchabu (113°07′ E, 40°59′ N, Inner Mongolia, north of China), Yuncheng (110°15′ E, 35°49′ N, Shanxi province, east of China), Shenyang (123°25′ E, 41°48′ N, Liaoning province, Northeast of China), Nanning (108.21 °E, 22.49 °N, Guangxi province, southwest of China), Suzhou (116.93 °E, 34.19 °N, Anhui province, south of China). The properties of soil and climate conditions in each site were shown in Appendix A. During the entire trial period, the average temperatures were from 22.4 to 28.1 °C. Moreover, the average rainfalls were from 152 to 2280 mm.

Two sites filled with yellow color indicate that the residue of CSO_2_ were higher than its LOQs in green soybean or soybean straw samples. Four sites filled with green color indicate that the residues of five analytes were all below than their LOQs in all samples.

To investigate the terminal residues of fomesafen, clomazone and clethodim in soybean, green soybean and soybean straw, 37% DOS of 721.5 g a.i./ha (grams active ingredient/hectare, the recommended dosage) was sprayed on the surface of soybean fields using a portable sprayer in the late stage of soybean seedling and the 2–5 leaf stage of weed for one time. Each experiment area was 50 m^2^. The soybean, green soybean and soybean straw samples (2 kg) were collected randomly at harvest with 79–80 of BBCH-identification (Biologische Bundesanstalt, Bundessortenamt and CHemical industry). All the collected samples were stored in dark at −20 °C for further analysis.

### 2.7. Residue Definition of the Three Herbicides

According to the Joint Meeting on Pesticide Residues (JMPR) report, the residue definition of fomesafen and clomazone for compliance with the MRL and for dietary risk assessment in plant and animal commodities were fomesafen and clomazone, respectively. The residue definition of clethodim in plant commodities intake involved clethodim and its metabolites CSO and CSO_2_. Total residues (*C_T_*), as the sum of clethodim and its metabolites CSO and CSO_2_, were calculated as follows [24,25]:(1)$$C_{T}=C_{CS}+C_{CSO}\frac{M_{CS}}{M_{CSO}}+C_{CSO2}\frac{M_{CS}}{M_{CSO2}}$$
where *C_CS_, C_CSO_* and *C_CSO_**_2_* represent the concentration of clethodim and its metabolites CSO and CSO_2_ residues, respectively, and *M_CS_*, *M_CSO_*, and *M_CSO_**_2_* represent molecular weights of clethodim (359.91 g/mol), CSO (375.91 g/mol), and CSO_2_ (391.91 g/mol), respectively.

### 2.8. Dietary Risk Assessment

For the safe application of fomesafen, clomazone, and clethodim in soybean fields, dietary exposure risk assessment was evaluated by risk quotients (RQ) method. RQ is calculated by dividing an exposure value by a toxicity end-point value [33]. RQ value >100 indicates an unacceptable risk for common consumers, while RQ value <100 presents an acceptable risk to human health. The chronic dietary exposure risk assessment was estimated by calculating *RQc* as follows:*NEDI* = ∑(*F_i_* × *SMTR_i_*)(2)
(3)$$RQc=\frac{NEDI}{ADI\times bw}\times 100\;\left(\%\right)$$
where *NEDI* (mg/kg bw) is the national estimated daily intake, *F_i_* (kg) is the reference food intake, *bw* (kg) is the average body weight, *STMR_i_* is supervised trials median residue for *F_i_* (if the STMR was not available, the corresponding MRL was used instead) and *ADI* (mg/kg bw) is the acceptable daily intake [33].

## 3. Results and Discussion

### 3.1. Optimization of Instrument Conditions

Two UPLC-ESI-MS/MS modes of negative and positive ions were compared for the five analytes of fomesafen, clomazone, clethodim, CSO, and CSO_2_. Compared with fomesafen, the other four analytes were more sensitive in the positive ion mode than in the negative one. Other determination parameters were shown in details in Table 1. In the UPLC-ESI-MS/MS analysis, the composition of the mobile phase significantly affects the peak shape, the retention time of the analytes and the signal response. Therefore, the composition, pH and flow rate of mobile phase were optimized. Suitable parameters were chosen to obtain the acceptable liquid chromatographic peak shape, intensity, retention time and sensitivity. Different mobile phase ratios (75:25, 50:50, and 25:75 of acetonitrile and water (*v:v*) with the addition of 0.2% formic acid or 10 mM ammonium acetate) were compared. After comparison with the different mobile phases, acetonitrile and 10 mM ammonium acetate water solution (75:25, *v:v*) were chosen with the retention time of fomesafen (0.75 min), clomazone (0.96 min), clethodim (1.31 min), CSO (0.78 min), and CSO_2_ (0.82 min), respectively.

### 3.2. Optimization of Sample Preparation

In previous studies, acetone or acid/phosphoric acid was used as a suitable extraction solvent for fomesafen analysis in juice or corn, while the recoveries ranged from 87.7% to 101.8% and the LOQs value were relatively large as 0.05 mg/kg in corn [8] and 5 mg/L in juice [29]. According to Hu and Hussan et al., acetonitrile was a good extract solvent for clomazone in soil or soybean matrix [14,15]. In addition, Ishimitsu and Wang etc. reported that acetone or acetonitrile were the proper extraction solvent of clethodim and its oxidation metabolites in several food samples [30,31].

Considering the results of previous studies and the high content of proteins and lipids of the soybean, green soyean and soybean straw matrices [14], acetone and acetonitrile were tested in our experiment. The results showed that the extraction efficiency of acetone was not satisfactory. Only fomesafen, CSO and CSO_2_ have been successfully recovered in the range of 70–120%, while the recoveries of clomazone and clethodim were lower than 70%. To our knowledge, clomazone and clethodim, with stronger polarity and water solubility (1100 and 5450 mg/L for clomazone and clethodim), were usually extracted with polar organic solvents such as acetonitrile [34,35]. Therefore, acetonitrile was selected to simultaneously extract for the five analytes in soybean, green soybean and soybean straw matrices. As a result, with the exception of fomesafen, a significant improvement of the recovery of the other four analytes was observed. Fomesafen is a weak acidic organic compound with pKa =2.83 [10]. When the pH of solvent is higher than the pKa, the compound appears in the dissociated form, which influences the extraction from food samples or the purification by dispersive solid-phase extraction (d-SPE) and the separation on reversed-phase column [5,36]. Crescenzi and co-workers reported that acidified extraction solvent with 0.2% trifluoroacetic acid was used to elute the acidic pesticides from organic sorbent such as GCB [37]. Hence, the extraction efficiency of 1% (*v:v*) formic acid in acetonitrile was used, showing a significant improvement. The extraction efficiencies of the five analytes in three matrices were acceptable in the rage of 71–116%.

Now, a lot of d-SPE sorbents, such as GCB, non-friable GCB (CarbonX), PSA, C18, florisil, MWCNTs, and ChloroFiltr etc., were used to purify the extracts of food, plant and other samples [23,37,38,39]. Based on their relative costs and adsorptive properties, PSA, C18, GCB, and MWCNTs were tested for the purification of the extracts of soybean, green soybean, and soybean straw. Recoveries of fomesafen, clomazone, clethodim, CSO, and CSO_2_ in soybean and green soybean matrix are shown in Appendix A using various d-SPE sorbents. From the figures, we could see the purification effect of PSA was not acceptable. The peak areas of fomesafen was decreased with the increase of the amount of PSA, and because fomesafen which was an acidic organic compound (pKa = 2.83) [10] was easily absorbed by PSA. As shown in Appendix A, only C18 was suitable to purify the extracts of soybean with recoveries ranging from 86% to 100%. As can be seen in Appendix A, when C18 was used to purify the extracts of green soybean, the recoveries were within the acceptable range of 80% to 100%. However, the samples still contained a large amount of chlorophyll (green color of samples) and the ME was higher than ± 30% for fomesafen, CSO and CSO_2_, respectively. Herein, in order to remove the chlorophyll from green soybean and soybean straw, 5 mg MWCNTs was added additionally. Finally, 5 mg MWCNTs, 50 mg C18, and 150 mg anhydrous MgSO_4_ were applied to purify the soybean extracts (1 mL) with perfect results, whilst satisfactory purifying effects could be obtained when 50 mg C18, 5.0 mg MWCNTs, and 150 mg MgSO_4_ were used to purify the green soybean and soybean straw extracts (1 mL).

### 3.3. Method Validation

To evaluate the linearity, matrix-matched standard calibrations were obtained by plotting concentrations via peak area. All regression data for fomesafen, clomazone, clethodim, CSO, and CSO_2_ in acetonitrile, soybean, green soybean and soybean straw are shown in Appendix A. The correlation coefficient (R^2^) of matrix-matched calibration was higher than 0.99 in the range of 0.005–1 mg/L for soybean, green soybean and soybean straw matrices, respectively (Appendix A).

To validate the accuracy and repeatability of the method, the recovery and RSDs were evaluated for the five analytes in the three matrices (soybean, green soybean, and soybean straw). Table 2 shows the recovery, corresponding RSDs and LOQs for the five analytes in the three matrices. The recoveries were in the range of 86–110% (RSDs ≤ 12.7) in soybean, 90–111% (RSDs ≤ 9.9) in green soybean, and 87–108% (RSDs ≤ 9.6) in soybean straw for all the analytes.

Meanwhile, the sensitivity was demonstrated to validate the established method. LOQ was determined as the lowest spiked concentration of target compounds in blank matrix. The LODs were defined as a signal-to-noise ratio of three with acceptable precision and accuracy.

In the above-mentioned conditions, the LOQs of the five analytes were 0.01 mg/kg in soybean, 0.01 mg/kg in green soybean, and 0.02 mg/kg in soybean straw, respectively. The LODs of the five analytes ranged from 0.018 to 0.125 μg/kg.

It was well known that a lot of interfering substances were co-extracted in extract procedure. These interfering substances can significantly affect the linearity, accuracy and sensitivity of analytes and potentially lead to erroneous quantification during detect by LC-MS or GC-MS. This ME can cause signal suppression or enhancement of the analytes on the LC-MS or GC-MS [40,41,42]. In this study, ME was evaluated for the five analytes in the three matrices and shown in Appendix A. The ME values of each matrix were ranged from −6.2% to 14%, which demonstrated that the matrix effect was acceptable and could be ignored.

The above findings demonstrate adequately the predominant performance in accuracy, reproducibility, and reliability of the proposed approach.

### 3.4. Terminal Residues of the Studied Herbicides in Soybean Ecosystems

The terminal residues of fomesafen, clomazone, clethodim parent and its two major metabolites in the soybean, green soybean, and soybean straw samples collected from Heilongjiang, Neimeng, Shanxi, Liaoning, Guangxi, and Anhui in 2018 are listed in Appendix A, respectively. It was easily observed that the terminal residues of CSO_2_ were the highest (0.015 mg/kg) in green soybean samples of Anhui, whilst the terminal residues of the five analytes were lower than LOQs (0.01 mg/kg) in all the other green soybean samples. In soybean straw samples, the terminal residues of CSO_2_ were as high as 0.029 mg/kg in Guangxi, and terminal residues of the other four analytes were lower than LOQs (0.02 mg/kg) in all soybean straw samples. In all soybean samples, the terminal residues of the five analytes were all lower than LOQs (0.01 mg/kg). These values were below the MRLs of 0.1 mg/kg established by China [43] or 2 mg/kg established by Japan [44].

According to the literatures [24,25], clethodim which was unstable at extreme pH values, temperature and UV light easily degraded to sulfone compounds such as CSO and CSO_2_. Most of these metabolites were more polar and more toxic than their parent compound which increased the probabilities of their adverse health effects [45]. It was well known that the persistence of pesticides in environment were influenced by temperature, pH, humidity, light radiation, soil type, and organic matter content and composition [24,25,45]. In this work, our results showed that clethodim easily degraded to sulfone compounds such as CSO and CSO_2_ in the open soybean ecosystem.

In previous studies, fomesafen residues were usually below their corresponding LOQs in tomato [5], earthworm [8], juice [29] and leeks [44] except 0.03 mg/kg in soil [5]. Clomazone residues were below its LOQs in soybean ecosystem [14,15]. The corresponding concentration of clomazone in environmental water was 0.2–0.9 µg/L. The residues of clethodim, CSO and CSO_2_ in rape plant and tobacco leaf were all blew their corresponding LOQs. These results all showed a lower residual level of the studied pesticides (Appendix A).

### 3.5. Dietary Risk Exposure Assessment for Three Pesticides

Chronic dietary exposure risk assessments of specific pesticides are needed for all registered crops that are subject to risk assessment [46]. The registered crops of fomesafen, clomazone, and clethodim in China and their corresponding MRLs recommended by different countries or international organizations are listed in Table 3. If the STMRs of three herbicides were not available, their corresponding MRLs were chosen to calculate NEDI values and to perform dietary exposure risk assessments.

The RQc value was calculated based on supervised field trial data and the results were also shown in Table 3, where the ADI values were 0.0025, 0.133 and 0.01 mg/kg bw for fomesafen, clomazone and clethodim, respectively, according to China Natiaonal Standard GB 2763-2019 [43]. Furthermore, 63 kg is the average body weight (bw) for an average Chinese adult [48]. The RQc values were 4.3 for fomesafen, 0.13 for clomazone and 19.3 for clethodim, respectively, which were significantly lower than 100. The results show that the human health risk of fomesafen, clomazone, and clethodim for soybean at the recommended dosage was negligible within the scope of ADI.

## 4. Conclusions

In this work, a simple, rapid, and effective analytical method based on the QuEChERS methodology and LC–MS/MS for the simultaneous determination of fomesafen, clomazone, clethodim, CSO, and CSO_2_ in soybean, green soybean, and soybean straw was developed and validated. Acetonitrile with 1% (*v:v*) formic acid, C18 and MWCNT were suitable extraction solvent and cleaners for simultaneous determination of five substances in soybean ecosystem with the recoveries ranged from 71% to 116% (RSD < 12.6%). The method met all international standards for pesticide residue analysis in terms of accuracy, sensitivity, linearity, correlation, and reproducibility. Further, this method was applied to the analysis of real field samples.

The field trials of the residue experiments were carried out at six sites in China in the year of 2018. Although the terminal residues of five substances were all below the MRLs established by China and Japan in soybean ecosystem at harvest time, the residues of CSO_2_ were appeared amount of 0.015 mg/kg in green soybean samples in Anhui, and 0.029 mg/kg in soybean straw samples in Guangxi, respectively. The dietary exposure risk assessments of the studied pesticides were performed in soybean and green soybean. The result implied that the dietary exposure risk of fomesafen, clomazone, and clethodim used in soybean and green soybean were acceptable to Chinese consumers. This study could provide guidance for the safe and reasonable use of three herbicides in soybean and green soybean.

## Figures and Tables

**Figure 1 ijerph-17-01951-f001:**
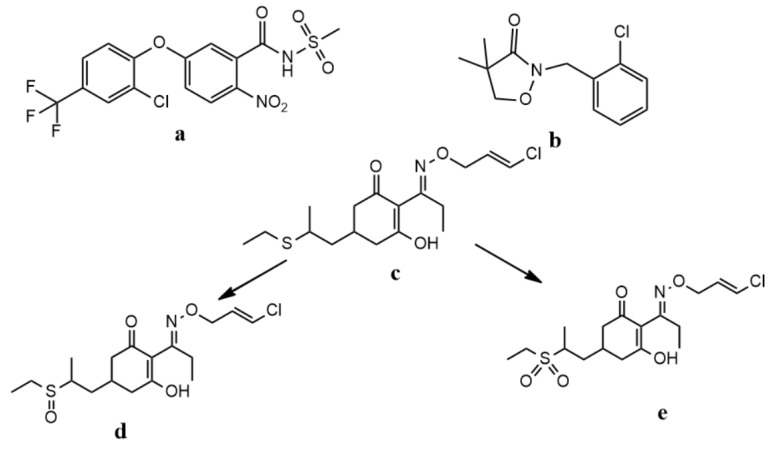
Chemical structures of fomesafen (**a**), clomazone (**b**), clethodim (**c**), CSO (**d**) and CSO_2_ (**e**).

**Figure 2 ijerph-17-01951-f002:**
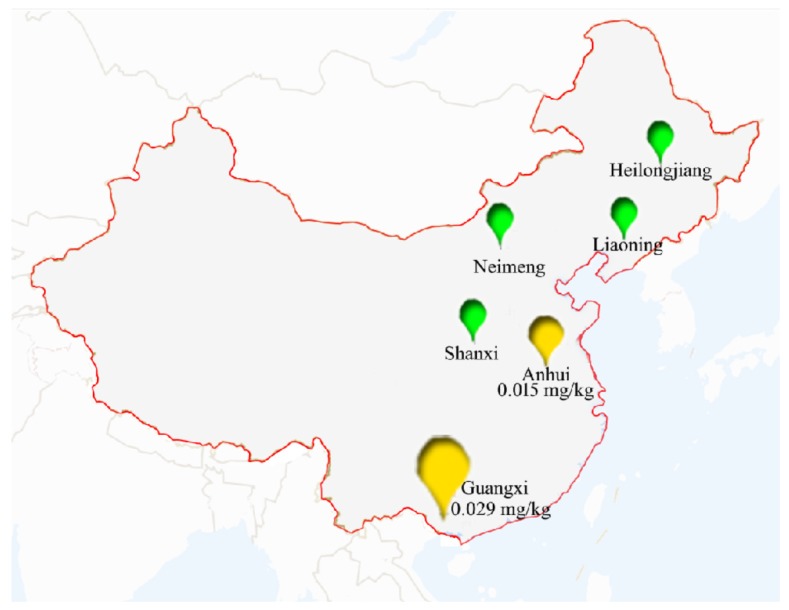
The Six sites of field trial in China in 2018.

**Table 1 ijerph-17-01951-t001:** UPLC-ESI-MS/MS parameters for determination of fomesafen, clomazone, clethodim, CSO and CSO_2_.

Compound (Molecular Formula)	Retention Time (min)	Transition m/z (Quantitative IonQualitative Ion)	Fragmentor Voltage (V)	Collision Energy (eV)	Polar
Fomesafen	0.75	437→286.1 437→315.9	150	20 25	Negative
Clomazone	0.96	240.1→125 240.1→89	85	21 55	Positive
Clethodim	1.31	360.1→164 360.1→206.1	85	17 15	Positive
CSO	0.78	376.1→206.1 376.1→164.1	100	15 25	Positive
CSO_2_	0.82	392.1→164.1 392.1→300.2	110	35 5	Positive

**Table 2 ijerph-17-01951-t002:** Recoveries, RSDs and LOQs of fomesafen, clomazone, clethodim, CSO and CSO_2_ in soybean, green soybean and soybean straw.

Matrix	Spiked Level (mg/kg)	Soybean		Green Soybean		Soybean Straw		LOQs (mg/kg)	LODs (μg/kg)
		Recoveries (%)	RSDs (%)	Recoveries (%)	RSDs (%)	Recoveries (%)	RSDs (%)		
Fomesafen	0.01 0.1 1.0	110 94 98	8.4 7.2 9.3	93 103 93	2.6 1.9 5.9	106 92 108	7.6 7.6 5.6	0.01	0.083
Clomazone	0.01 0.1 1.0	105 94 101	8.7 2.9 3.2	100 90 94	9.9 2.6 3.9	91 97 87	9.2 3.3 1.8	0.01	0.018
Clethodim	0.01 0.1 1.0	108 100 93	6.4 8.8 6.7	105 91 96	7.9 1.6 1.9	99 99 90	9.6 3.0 3.3	0.01	0.042
CSO	0.01 0.1 1.0	92 91 86	3.4 4.9 9.7	97 90 98	5.9 1.8 1.6	92 100 98	1.5 2.1 1.6	0.01	0.077
CSO_2_	0.01 0.1 1.0	89 86 91	12.7 9.8 9.8	111 93 94	1.9 3.8 3.0	89 98 88	4.3 1.8 3.8	0.01	0.125

**Table 3 ijerph-17-01951-t003:** Chronic dietary risk assessment of fomesafen, clomazone and clethodim based on the Chinese dietary pattern.

Herbicides	Crops	Food Classification	Fi (kg)	STMR_i_ (mg/kg)	*Sources* [43,47]	NEDI (mg/kg bw)	ADI (mg/kg bw)	RQc
Fomesafen	Soybean	Vegetable oil	0.016	0.01	STMR	0.00016	0.0025	
	Peanut	Vegetable oil	0.0327	0.2	China	0.00654		4.3
Clomazone	Soybean	Vegetable oil	0.016	0.01	STMR	0.00016	0.133	
	Potato	Tubers	0.0495	0.02	China	0.00099		
	Sugarcane	Sugar, starch	0.0044	0.1	China	0.00044		
	Pumpkin	Light vegetables	0.0915	0.05	China	0.004575		
	Rice	Rice and its products	0.2399	0.02	China	0.004798		0.13
Clethodim	Soybean	Vegetable oil	0.016	0.03	STMR	0.00048	0.01	
	Garlic	Sauce	0.009	0.5	European Union	0.0045		
	Tomato	Dark vegetables	0.0915	1	European Union	0.0915		
	Sugarbeet	Sugar, starch	0.0044	0.1	European Union	0.00044		
	Potato	Tubers	0.0495	0.5	European Union	0.02475		19.3

**Note**. *NEDI* (mg/kg bw) is the national estimated daily intake, *F_i_* (kg) is the reference food intake, *STMR_i_* is supervised trials median residue for *F_i_* (if the STMR is not available, the corresponding MRL is used instead) and *ADI* (mg/kg bw) is the acceptable daily intake, RQ is chronic dietary exposure risk. RQ value >100 indicates an unacceptable risk for common consumers. *Source* presents the source of *STMR_i_*.

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
