# Peer review of "Simultaneous Analysis and Dietary Exposure Risk Assessment of Fomesafen, Clomazone, Clethodim and Its Two Metabolites in Soybean Ecosystem"

_ijerph, 2020, doi:10.3390/ijerph17061951_

Round 1
Reviewer 1 Report
I suggest the Authors improve the explanations about the several figures and tables, as well as, the theoretical background behind.
The conclusions section is very week. After several work it is not fair for the study to present some lines as conclusions.
Author Response
Dear reviewer,
Thank you very much for your professional comments and thoughtful suggestions about our paper (Manuscript ID: ijerph-738062) submitted to IJERPH. We have revised the manuscript according to your valuable comments and carefully proof-read the manuscript. All changes have been made to the text in red colour. We hope the revised manuscript can meet the magazine’s standard. You will find our point-by-point responses to the reviewers’ comments in “Author to respond reviewer1.docx”.
Best regards,
Yours sincerely
Jiye Hu

Reviewer 2 Report
In my judgment, after review carefully the manuscript, the changes are adequate and the authors have improved it properly. The confusing issues are also clearer. Currently, the article includes all the necessary information for a proper understanding of the work and results are of interest.
Author Response
Dear reviewer,
Thank you very much for your professional comments and thoughtful suggestions about our paper (Manuscript ID: ijerph-738062) submitted to IJERPH.
Best regards,
Yours sincerely
Jiye Hu
This manuscript is a resubmission of an earlier submission. The following is a list of the peer review reports and author responses from that submission.
Round 1
Reviewer 1 Report
From bio-analytical points of view, this work is simply a routine application of state-of-art instrumentation UPLC-ESI-MS/MS. If the presented data didn't provide any information of scientific significance in the field of environmental science, the reviewer will be against of the publication of this manuscript.
The interested concentration range 10 - 1000 mg/kg, equivalent to 10 to 1000 ppm, is obvious higher than the detection sensitivity of triple quadrupole analyzer. The reviewer didn't recognize any importance to investigate the development works of these methods for the sake of scientific publications.
Author Response
Thank you so much for your comments on our manuscript.
In our opinion, our work had three highlights.
Firstly and secondly, it was not only a LC-MS analysis of these analytes which had scarcely reported but also a field analysis of terminal residue as well as dissipation. Besides, it was a simultaneous determination of these analytes which was also a difficult point for the analysis of these analytes. There was a distance between the LC-MS method establishment and our work, considering the simultaneous determination, QuChERS sample preparation and field anlysis. Our QuChERS sample preparation provided an optimum method for the determination of fomesafen, clomazone, clethodim and its metabolites, (CSO and CSO2) in soybean, green soybean and soybean straw samples. Finally, it analyzed the dietary risk assessment of those targets which had not reported previously. Thus, it was not only a LC-MS determination of those analytes. In addition, our work quantitative residues 0.01-2 mg/kg with a LOQ of 0.01 mg/kg, which was satisfactory for a LC-MS.
Thank you so much for your help. If there are any problems, please let me know. Looking forward to hearing from you at your earliest convenience.
Best regards.
Sincerely yours,
Reviewer 2 Report
About the submission with the title "Simultaneous analysis and dietary exposure risk assessment of fomesafen, clomazone, clethodim and its two metabolites in soybean ecosystem" I have the following comments:
I suggest the Authors improve the abstract with the following points: motivations, objectives, methodologies and main insights. About the main insights, please, present them in a more clear and understandable way. Do not forget that in the abstract the general readers begin with the first contact with you research and if you are too much technical some messages may be not understandable. For exanple, what means "The mean recoveries ranged from 71% to 116% with relative standard deviations (RSDs) less than 12.6%."? What means? The remain abstract was presented in a similar not well understandable way.
In the introduction section you need to improve the literature review, because the are many studies related with these topics do not cited.
In the section two you need to support your several options with other studies already published. If not it seems you made some decisions ad-hoc.
In the section 3 you need, also, to compare your results with others from other studies and present a critical perspective.
The conclusions section is very week for a scientific paper. Please, improve significantly it.
Author Response
Dear Reviwer,
Thank you so much for your comments on our manuscript. According to the suggestions from you, we have revised the manuscript carefully. The changes and corrections were in red in the revised manuscript. Enclosed please find our responses to reviewers’ comments as well as the revised manuscript. We thanked you from the heart whose questions indicated that they are very professional and familiar with our work.
Thank you so much for your help. If there are any problems, please let me know. Looking forward to hearing from you at your earliest convenience.
Best regards.
Sincerely yours,

Reviewer 3 Report
This work develops a method for the simultaneous determination of several pesticides commonly present in soybean, green soybean and soybean straw. The method is based on QuEChERS sample preparation and UPLC-MS/MS. The manuscript fits within the scope of the journal and there is sufficient information present to replicate the research. The materials and methods have been adequately described and the analysis of the technical aspects is correct. Conclusions are correct. However, this reviewer has identified several issues that impede publication of the manuscript in the current form:
The English need to be improved. The discussion of results is poor. A simple description of results, without comparing them with those obtained in other works, is not enough. References should be revised deeply, because they are not suitable or they are not assigned properly. For example: > reference 14 of line 56 is for soil and rape, not for aqueous media. It should be replaced for example by “Sci. Tot. Environ. 2018, 615, 643-651”, “J. Agric. Food Chem. 2010, 58, 3068-3076”, etc. > reference 22 of line 194 uses SPE for sample preparation, not QuEChERS. It should be replaced for example by “Molecules 2018, 23, 2009”, “Int. J. Environ. Anal. Chem. 2013, 93, 1566-1584”, etc. > reference 23 of line 194 does not analyse soil or paddy water. It should be replaced for example by “J. Anal. Chem. 2016, 71, 508-512”, “Bull. Environ. Contam. Toxicol. 2011, 86, 18-22”, etc.
Taking into account the comments to the authors, I consider that major requirements are still needed before the manuscript could be accepted in this journal.
Author Response

(The authors gave the same response as above.)

Round 2
Reviewer 1 Report
Assuming the authors would like to highlight the significance of developing QuECHERS methods under open field conditions to quantify pesticides applied on soybeans, the reviewer suggests to determine the pesticides using portable GC or biosensors. The expensive UPLC-MS/MS is only accessible in central lab. There is no point to develop sample preparation methods under open field conditions and then to quantify these samples in central lab.
Again, the LOD at the level of tens of ng/mL is nothing special using LC-MS/MS. The bioanalytical merits of this study are difficult to justify.
Reviewer 2 Report
I suggest the Authors consider deeper my suggestions from my first report and improve (more) the submission. For example, the conclusions section need to be significantly extended with the main insights from the research.
Reviewer 3 Report
In my judgment, after review carefully the manuscript, the changes are adequate and the authors have improved it properly. The confusing issues are also clearer. Currently, the article includes all the necessary information for a proper understanding of the work and results are of interest.